# *Cerianthus lloydii* (Ceriantharia: Anthozoa: Cnidaria): New Status and New Perspectives [note 1]

**DOI:** 10.3390/biology12091167

**Published:** 2023-08-24

**Authors:** Tina N. Molodtsova, Viktoria N. Moskalenko, Elizabeth V. Lipukhin, Tatiana I. Antokhina, Marina S. Ananeva, Ulyana V. Simakova

**Affiliations:** 1Shirshov Institute of Oceanology RAS, 36 Nakhimovsky Prospect, Moscow 117218, Russia; 2Severtsov Institute of Ecology and Evolution RAS, 33 Leninski Prospect, Moscow 119071, Russia

**Keywords:** ceriantharia, *Cerianthus*, *Synarachnactis*, geographic distribution, Arctic, North Atlantic, North Pacific, Black Sea fauna, taxonomy

## Abstract

**Simple Summary:**

Subclass Ceriantharia encompasses marine anemone-like organisms with complete bilateral symmetry. In modern phylogenetic reconstructions, Ceriantharia is considered an ancient group that is sister to all other anthozoans. Only a handful of tube anemones have been reported from the Arctic, including widely distributed North Sea cerianthid *Cerianthus lloydii* Gosse, 1859, which has also been documented in temperate Atlantic and Pacific waters. The integrity of *C. lloydii* as a species has been questioned in the literature. To test this hypothesis, we performed a molecular study of *C. lloydii* from several geographically distant localities using 18S and COI genes. Our data combined with data from public databases show that *C. lloydii* forms a single genus-level group divided into three distinctive subclades: (1) Northern Europe, the Black and the Barents seas; (2) the Russian and Canadian Arctic and the Labrador Sea; and (3) published sequences of a species determined as *Pachycerianthus borealis* (Verrill, 1873) from Newfoundland. As all subclades of *C. lloydii* are only distantly related to the genus *Cerianthus* Delle Chiaje, 1841, we propose to resurrect the genus *Synarachnactis* Carlgren, 1924, and describe a new family to accommodate them. Our findings contribute to the understanding of the marine fauna evolution in the Arctic Basin and the Black Sea.

**Abstract:**

Subclass Ceriantharia is a well-defined and probably ancient group of marine benthic organisms renowned for their bilateral symmetry, which is reflected in the arrangement of tentacles and mesenteries. Four species of Ceriantharia have been reported in the Arctic, including *Cerianthus lloydii* Gosse, 1859, also known from the Northern Atlantic and Northern Pacific. The integrity of this species was questioned in the literature, so we performed a molecular study of *C. lloydii* from several geographically distant locations using 18S and COI genes. The phylogenetic reconstructions show that specimens of *C. lloydii* form a single group with high support (>0.98), subdivided into distinctive clades: (1) specimens from Northern Europe, the Black and Barents seas, and (2) specimens from the White, Kara, Laptev, and Bering seas and also the Canadian Arctic and the Labrador Sea available via the BOLD database. There are several BOLD COI sequences of *Pachycerianthus borealis* (Verrill, 1873), which form a third clade of the *C. lloydii* group, sister to the European and Arctic clades. Based on low similarity (COI 86–87%) between *C. lloydii* and the type species of the genus *Cerianthus* Delle Chiaje, 1841—*C. membranaceus* (Gmelin, 1791), we propose a new status for the genus *Synarachnactis* Carlgren, 1924, and a new family Synarachnactidae to accommodate *C. lloydii*.

## 1. Introduction 

Ceriantharia, also known as tube anemones, is a well-defined [1] and relatively ancient [2] group of marine benthic organisms. Ceriantharians are solitary anemone-like anthozoans known from the intertidal zone to the abyssal depths of all oceans [3]. Tube anemones have a rather uniform appearance, with an elongated column and flattened oral disk with two crowns of tentacles [3,4]. Tube anemones ordinarily inhabit long fibrous tubes that give their vernacular name [1,4]. These tubes are composed of the special cnidae type and encrusted with surrounding particles [5,6]. Ceriantharians are famous for their bilateral symmetry manifested in the arrangement of the tentacles and mesenteries, which allows to use Ceriantharia in comparative zoology as a model group to explain the evolution of metazoans and their shift to Bilateria [7]. Currently, in accordance with molecular studies, Ceriantharia is considered to be one of three subclasses of the class Anthozoa of the Phylum Cnidaria [3,8].

Most ceriantharian species were described from the shallow tropics. The fauna of ceriantharians in the high latitudes is less diverse [4,9]. Until now, four species of Ceriantharia—all from the family Cerianthidae—have been reported from the high latitudes of the Northern Hemisphere [9,10]. These are *Cerianthus lloydii* Gosse, 1859, in the Arctic seas and adjacent areas of the Pacific and Atlantic Oceans [4,9,11,12]; *C. roulei* Carlgren, 1912, from the Svalbard area [9,13]; *C. vogti* Danielssen, 1890, from the Norwegian and Laptev seas [4,12,13,14], and *Pachycerianthus borealis* (Verrill. 1872) reported from the Canadian Arctic [10]. Among these four species *C. lloydii* is considered to have one of the widest distributions [4,9,11]. That may be explained by the long-living planktonic larvae [15]. However, due to disjointed records in the Pacific Ocean, the integrity of this species has been debated [4]. In addition, the position of *C. lloydii* within the genus Cerianthus Delle Chiaje, 1841, was challenged based on molecular data [3,4].

The main aim of our work is to test the hypothesis of non-integrity of the North Sea cerianthid [4] based on a molecular study of specimens from several geographically distant localities including the Arctic, Atlantic, and Pacific oceans using two molecular markers (18S and COI) and to clarify the position of *C. lloydii* within the subclass Ceriantharia.

## 2. Materials and Methods

### 2.1. Taxon Sampling and Identification

A total of 28 specimens of Ceriantharia from the collections of the P.P. Shirshov Institute of Oceanology RAS, Moscow, Russia (IORAS, 25 specimens); the National Museum of Natural History, Washington, DC, USA (USNM, 2 specimens); and the Florida Museum of Natural History (FLMNH, 1 specimen) were used in the present study. Detailed information on the museum catalog numbers and geographical localities of each specimen can be found in the Appendix A. Only specimens with vouchers available for morphological assessment and a subsample preserved for genetic studies were used in the present work. Determination of tube anemones was validated using routine examination of arrangement and morphology of tentacles and mesenteries under a dissecting microscope Olympus SZX7 at ×5–28 and a study of cnidae squash preparations under a light microscope Olympus BX51 at ×1000 as described in the literature [3,16,17]. We use terminology as summarized in the supplementary Glossary of Ferero Mejia and co-authors [3].

### 2.2. CO1 and 18S Sequence Acquisition

Genomic DNA was extracted from 96% ethanol fixed fragments. The samples were exsiccated at 50 °C before lysis. DNA was extracted using DNeasy Blood & Tissue Kit (Qiagen™, Hilden, Germany) according to the manufacturer’s recommendations. In some cases, extra clean-up was performed using the Genomic DNA Clean & Concentrator kit, Zymo (Irvine, CA, USA). The primer pairs and annealing temperatures used in this study are listed in Table 1. A pre-made PCR mix (ScreenMix-HS) from Evrogene™, Moscow, Russia (ScreenMix-HS, 0.5 μM of each primer and 1 μL of DNA), was used for the amplification. The resulting PCR product was visualized in a 2% agarose gel, purified using ethanol-ammonium acetate precipitation, and sequenced with the same primers using ABI PRISM^®^ BigDye™ Terminator v. 3.1 on Applied Biosystems (Foster City, CA, USA) DNA Analyzer 3500 ABI.

Chromatograms were processed using Codon Code Aligner 9.0.1 (Codon Code Corporation, Centerville, MA, USA). The resulting sequences after primer trimming were deposited in the GeneBank [22] (Appendix A). All sequences for the protein-coding region were checked for a stop-codon presence using TranslatorX [23]. Additional Ceriantharia sequences representing all available ceriantharian benthic genera were obtained from GeneBank and BOLD databases (Appendix A). Type species of each genus were preferred where available. All existing sequences from genera *Ceriantheopsis*, *Botrucnidifer*, and *Cerianthus lloydii* and *Cerianthus borealis* (clade A *sensu* Forero Meija and co-authors, for sake of comparability, hereafter we use the same clade names as applied in [3]), were assessed. Black coral *Leiopathes* sp. (Hexacorallia Antipatharia Leiopathidae) was chosen as an outgroup. Fasta files were aligned using the MAFFT 7.308 [24] with the manual check and correction. KHA043-14, KHA043-14, and KHA045-14 (all BOLD) were excluded from analysis because of misidentification (after running BLAST, they were found 100% identical to *Metridium senile* (Linnaeus, 1761) (Actiniaria)). For phylogenetic reconstructions, we used the alignments of both genes independently (for 18S gene—with BMGE [25] processing) and concatenation of COI and 18S. With a few exceptions (*Ceriantheomorphe brasiliensis*, *Ceriantheopsis americana*, and *Isarachnanthus* spp., see Appendix A), sequences taken from the same specimens were chosen for concatenated trees.

### 2.3. Phylogenetic Analyses

To obtain phylogenetic reconstructions with Bayesian inference, we used the MrBayes 3.2 [26] (BI, chain length 10,000,000, 3 heated and 1 cold chains, hcTemp 0.1, subsampling frequency 1000, 4 independent runs, first 10% of samples were discarded). To choose the model of evolution, the PartitionFinder 2 [27,28] was applied with the 4 (concatenated) and 3 (protein-coding only) initial partitions and use of a greedy algorithm with Bayesian information criterion (BIC) [29].

We also employed the maximum likelihood (ML) method implemented in the IQ-TREE 1.6.12 software [30]. To assess branch support, ultrafast bootstrap [31] approximation (UFboot) and the SH-like approximate likelihood ratio test (SH-aLRT) [27] with 10,000 bootstrap replicates were used). The substitution models were chosen according to the Bayesian information criterion (BIC) with the help of the ModelFinder [32] implemented in the IQ-TREE software. Obtained phylogenetic trees were visualized with the help of ITOL v.6.7.5 [33].

The number of base differences per sequence from averaging over all sequence pairs between and within groups was calculated using MEGA X [34]. The rate variation among sites was modeled with a gamma distribution (shape parameter = 4). The 18S analysis involved 46 nucleotide sequences and 1623 positions in the final dataset. For the COI, the 52 nucleotide sequences with 651 positions were used. All ambiguous positions were removed for each sequence pair (pairwise deletion option).

## 3. Results

### 3.1. Phylogenetic Analyses

A total of 51 new sequences (27—COI, 24—18S) belonging to nine species of Ceriantharia were generated in our study (Appendix A). We combined newly generated sequences with previously published data for Ceriantharia available in GenBank and BOLD (Appendix A), thus all hitherto known benthic genera of Ceriantharia were included in our dataset. Calculated estimates of evolutionary divergence over sequence pairs between and within clades are presented in Appendix A.

Both ML and BI phylogenetic reconstruction based on concatenated 18S and COI alignments show the same topology and the main clades supports (Figure 1; Appendix A). 

Rooted to *Leiopathes* sp., our trees show well-supported clades, with Ceriantharia gen. nov sp. Nov. USNM 1457395 being a sister group to all hitherto sequenced species of Ceriantharia (for distances, see Appendix A). Other Ceriantharia are divided into two well-supported groups of clades (SH-aLRT > 80%, UFboot > 95%, (Figure 1; Appendix A). Clade A includes three independent clades A1–A3, corresponding to genera *Ceriantheopsis* (A1; two species including type species of the genus, *Ceriantheopsis americana* (Agassiz in Verrill, 1864)), *Botrucnidifer* (A2; three new species under description), and a species group previously reported as *Cerianthus lloydii* (A3). The second group consists of clades B-D, which correspond well to those previously reported [3]. 

For the first time, the genus *Arachnanthus* Carlgren, 1912 (Arachnactidae), represented by *A. sarsi* Cargren, 1912, was included in genetic studies. *Arachnanthus sarsi* nests well in Arachnactidae as a sister group to all *Isarachnanthus* species included in the analysis. *Pachycerianthus solitaruis* (Rapp, 1829) is found within other *Pachycerianthus* spp. in clade C, which also includes *Cerianthus* cf. *mortenseni*. The family Botrucnidiferidae is polyphyletic, and each genus of this family included in the study belongs to a well-supported subclade in clades A (*Botrucnidifer*) and B (*Botruanthus*). Clade B comprises *Botruanthus*, *Cerianthomorphe*, and *Cerianthus*, represented by type species of each genus. Thus, members of the genus *Cerianthus* included in our dataset are distributed between clades A, B, and C. 

Comparing the results of the 18S- and COI concatenation-based phylogenetic reconstruction with the individual analysis for each gene, some contradictions can be found. The position of Ceriantharia gen. nov. sp. Nov. USNM 1457395 as sister to all other Ceriantharia is not stable. In the COI tree, it is placed within clade A, while the *Ceriantheopsis* (A1) becomes a sister group to all other species with low to moderate supports (SH-aLRT < 80%, UFboot < 95%, Appendix A). Although the bipartition of previously known Ceriantharia (clades A and B–D) is preserved in the 18S-based tree, the supports of these groups are weak (SH-aLRT < 80%, UFboot < 95%, Appendix A). Thus, clade A splits into three lineages with unclear relationships. The resolution of 18S is not sufficient to resolve the relations between clades B-D as well. Only Arachnactidae (clade D) remains certain (SH-aLRT and UFboot = 100%). *Botruantus benedeni* subclade and *Cerianthus membranaceus*—*Cerianthus* sp. RS03 subclade also remain but with weak supports.

We compared COI sequences of the A3 subclade from the Arctic Basin, the Black Sea, Northern Europe, and the Bering Sea with the North Atlantic and Arctic ceriantharians from GeneBank and BOLD. Three moderately to weakly supported lineages were revealed by both ML (Figure 2) and BI (Appendix A) phylogenetic reconstructions. The first lineage consists of specimens from Northern Europe, the Black Sea, and the Barents Sea including previously published samples from Kattegat. Samples from the White, Kara, Laptev, and Bering seas belong to the second lineage, genetically different from *Cerianthus lloydii* from Northern Europe and the Barents and Black seas. The same lineage includes sequences obtained from BOLD specimens from the Canadian Arctic. Several publicly available sequences from the Gulf of St. Lawrence (Northwest Atlantic; Newfoundland) identified in BOLD as *Pachycerianthus borealis* form a third lineage in the A3 (Figure 2; Appendix A), which is sister to both the European and Arctic lineages. Although the genetic distances (Appendix A) between lineages are low (0.01–0.02), the distances within each of the three subclades are about ten times lower (0.001). Each subclade demonstrates specific distinguishing nucleotides in all variable positions.

### 3.2. Species in A3 Subclade

The structure and arrangement of mesenteries in studied specimens from both lineages of A3 subclade were very similar and corresponded well to those of *Cerianthus lloydii* [9,11,13]. Few differences could be explained by the relative size of specimens, with specimens from the Arctic-Pacific clade normally being larger than those from the European seas and the Black Sea. At least some specimens in the studied collections were easy to distinguish from each other based on the underwater photographs (Figure 3): specimens from the European lineage often have translucent annuli at the base of marginal tentacles (Figure 3a), and specimens of the Arctic lineage commonly have lighter tentacles without visible annuli (Figure 3b). However, several darker and lighter morphs are usually present in the same populations. In addition, specimens from the Arctic clade have larger polyps with longer and thicker tentacles and a higher position above the seafloor (compare Figure 3c,d). We were also able to distinguish two species using squash preparations from labial and marginal tentacles. Normally, specimens of the Arctic-Pacific clade had larger b-rhabdoids (up to 33 μm in marginal tentacles and up to 47 μm in labials vs. 26 μm (marginals) and 30 μm (labials) in the European clade, see Figure 4). These characteristics are also distinctive for the syntypes of *Cerianthus roulei* Carlgren, 1912 (MOM Inv-20725, TNM, personal observations). This species was originally reported as *Cerianthus lloydii* from Icefjord, Svalbard, by Albert 1er of Monaco Expedition [35] and subsequently given the name *C. roulei* due to unusually illustrated polyps [13]. In the course of visual observations in the White and Bering seas (T.I.A., personal observations), no decapods associated with tube anemones were observed. Neither were they observed in video footage from Severnaya Zemlya (materials of the expedition “Open Ocean: Archipelagos of the Arctic—2019. Severnaya Zemlya”). Re-description of these two species and amended diagnoses will be published separately along with a more detailed map of geographical distribution based on morphological characters.

## 4. Discussion

### 4.1. Phylogeny of Ceriantharia

Our dataset covers all eight hitherto described benthic genera of tube anemones and 17 nominal and six undescribed species of 54 nominal hitherto known species [36] (~31.5% of total nominal species count), with 10 species in the clade A *sensu* Forero Mejia and coauthors [3] (Figure 1; Appendix A; Appendix A). 

The position of USNM 1457395 at concatenated 18S COI trees (Figure 1; Appendix A) as a sister group to all previously reported clades of Ceriantharia suggests it to be a highly divergent taxon, probably of a rank of a new family or higher. This hypothesis is supported by the complete absence of the characteristic cerianthid tube in this new species and its bathypelagic mode of life. Additional studies of USNM 1457395 using several nuclear loci are currently underway to describe its phylogenetic position within Ceriantharia. The description of this unusual taxon is planned to be published separately.

Our phylogenetic reconstructions demonstrate strong agreement with the results obtained by Forero Mejia et al. [3], with the exception that clade A is a very diverse entity. The three subclades composing clade A (two subclades represented by the genera *Ceriantheopsis* and *Botrucnidifer*, respectively, and the third subclade formed by *Cerianthus lloydii* species complex) constitute a highly supported lineage distinct from Cerianthidae and Arachnactidae. Although the interrelationship of *Ceriantheopsis, Botrucnidifer*, and *Cerianthus lloydii* species complex is well supported in the concatenated phylogenetic tree, the contradiction between reconstructions based on separate genes indicates the need for further study of clade A with more representatives of these genera and the type species of *Ceriantheopsis* and *Botrucnidifer*.

Apparently, all the subclades in Clade A have to be considered as family-level groups. However, in the absence of nuclear genetic markers for the type species of the genus *Ceriantheopsis*, we consider any nomenclature acts with this subclade as preliminary. As was shown recently by Quattrini and co-authors [37], there are significant discordances between nuclear and mitochondrial trees that may affect observed phylogeny if only one type of the markers is used. Botrucnidiferidae Carlgren 1912 is a long-time established family, and its polyphyly was already discussed by Forero Meija and coauthors [3], but absence of any genetic data for the type species of the genus *Botrucnidifer* makes it difficult to understand the relationships of species within the subclade. Species determined as *Botrucnidifer* spp. in our work and those published in Forero Meija and co-authors [3] probably belong to two different genera. Nevertheless, the clade A3 comprising the *Cerianthus lloydii* complex can be considered as a new family.

### 4.2. Taxonomic Implications

#### 4.2.1. Synarachnactis Carlgren 1924 (Amended)

From our dataset (Figure 1, Appendix A), it is possible to see that specimens originally determined as *Cerianthus lloydii* [11] are distant from the type species of the genus *Cerianthus* (*Cerianthus membranaceus*, clade B) and form a well-supported clade sister to genus *Botrucnidifer.*

To resolve polyphyly of the genus *Cerianthus*, for the last group, we formally propose to resurrect the genus *Synarachnactis* Carlgren, 1924, with the type species *Synarachnactis bournei* (Fowler, 1897) [37] established for cerianthid larvae subsequently recognized as the larval stage and junior synonym of *Cerianthus lloydii*. The genus name *Synarachnactis* Carlgren, 1924, will be hereafter used for this group. 

**Diagnosis** of the genus *Synarachnactis* is proposed as follows.

Tube inhabiting ceriantharians. Marginal and labial tentacles arranged in three to four pseudocycles. Directive labial tentacle absent. Siphonoglyph small, hyposulcus feebly developed. Directive mesenteries very short. Protomesenteries 2 long, fertile, reaching aboral pole and provided with short distinctive ciliated tract without craspedonemes, very short cnido-glandular tract, and very long craspedion region. Protomesenteries 3 rather short with ciliated tract without craspedonemes and well-developed cnido-glandular tract. Arrangement of metamesenteries in each quartette MBMb, where M mesenteries reach the aboral pole and have the same structure as protomesenteries 2, and B and b mesenteries have the same structure as protomesenteries 3. B-mesenteries are longer than b in each quartette with a better developed ciliated tract. All metamesenteries gradually diminishing in length towards the multiplication chamber. Planktonic larvae reported in at least some *Synarachnactis* species. 

The genus *Synarachnactis* can be distinguished from known genera of Ceriantharia by (1) complete absence of craspedonemes of the ciliated tract (from *Cerianthus, Botruanthus, Pachycerianthus, Ceriantheopsis*), (2) absence of cnidorages (from *Botruanthus, Botrucnidifer*) and (3) absence of acontioids (from *Arachnanthus, Isarachnanthus*). From all hitherto known genera, *Synarachnactis* can be distinguished by arrangement of metamesenteries in each quartette: *MBMb*. Species recognized: *Cerianthus lloydii* Gosse, 1859, and *Cerianthus roulei* Carlgren, 1912. 

#### 4.2.2. Synarachnactidae fam. nov. Molodtsova and Simakova, 2023 

[LSID urn:lsid:zoobank.org:act:7478FA55-8DF8-4A1C-A507-EB9DB0F00568]

Based on our phylogenetic reconstruction, we also formally propose a new family Synarachnactidae Molodtsova and Simakova, 2023, to accommodate *Synarachnactis* spp., with the following **diagnosis**: 

Tube-inhabiting ceriantharians without craspedonemes in the region of the ciliated tract, with no acontioids or botrucnids at any part of the mesenterial filament. Protomesenteries 2 long, fertile, may reach the aboral pole. Metamesenteries arranged MBMb in each quartette. 

The only genus in the family is *Synarachnactis* Carlgren, 1924.

### 4.3. Geographic Distribution of the Genus Synarachnactis

Observed distributional patterns with two species reported from the Atlantic (*Synarachnactis lloydii* in the North-East Atlantic and *Synarachnactis* indet. from Newfoundland) and one (*S. roulei*) from the Arctic and Pacific (Figure 5) most likely reflect repeated invasion events of Pacific fauna into the Arctic [38,39]. We can hardly speculate about the time frame of these events with such limited genetic data from the Pacific and North-West Atlantic and limited records on *Synarachnactis* species distribution. Characteristic *Synarachnactis* larva, *Synarachnactis brachiolata* (A. Agassiz, 1863), was reported from plankton of the Western Atlantic off of Boston, USA [40]. Later, the larvae was assigned to *Pachycerianthus borealis* (Verrill, 1872); nevertheless, an adult cerianthid determined as *Cerianthus lloydii,* that may represent one of three *Synarachnactis* revealed in our study has also been reported from a nearby locality [9]. *Synarachnactis brachiolata* may represent the third species of the genus or may turn out to be a synonym of one of the former two. Genetic studies of *Synarachnactis* from the Pacific, for example from the shallow-water hydrothermal ecosystem in the Kurile Islands, the Sea of Japan, and adjacent areas [9,11], are crucial for determining the timing of *Synarachnactis* diversification and its pathways in the Arctic and may also contribute to further understanding of the evolution of marine fauna in the Arctic Basin.

The presence of *Synarachnactis lloydii* in the Black Sea is surprising, but it should not be speculated as a recent invasion. In the Black Sea, the species was collected in a very characteristic biotope—at a low shelf, often in association with bivalve *Modiolula phaseolina* (Philippi, 1844) [41]. This type of community has been reported since the 1950s all around the Black Sea, usually associated with temporal or permanent oxygen depletion zones [42,43,44,45,46,47]. Ceriantharians in these communities were usually reported as *Cerianthus vestitus* (Forbes, 1843) or lately as *Pachycerianthus solitarius* (Rapp, 1829). *Cerianthus vestitus* was originally described by Edward Forbes from the shallow sandy habitats of the Northern Paros (the Aegean Sea) [48] as a species of the genus *Edwardsia* with two rings of tentacles and a characteristic protective tube [49]. No morphological data were provided in the description, and the original material is probably lost. Later, this species was re-considered in Ceriantharia as a possible synonym of *Cerianthus membranaceus* [50] or *Pachycerianthus solitarius* [51]. The first published identification of the Black Sea ceriantharian as *C. vestitus* is apparently dated 1900 or earlier and is based on collections from the Russian Black Sea explorations 1890–1891 [52,53]. At this time, features of the inner morphology of anemones were only rarely reported or used for identification. This early determination was probably based on the minute size of the Black Sea ceriantharian and its few tentacles (Forbes [48] mentions 32 outer tentacles). It is very likely that in subsequent publications, the Black Sea ceriantharian was reported mainly based on that first determination and its presence within a temporal or permanent oxygen depletion zone. Our work is the first detailed examination of the Black Sea ceriantharian, in terms of both morphology and genetics.

To eliminate any probable misidentification, we compared genetic data of ceriantharians from the Black Sea with those for both senior synonyms proposed for *C. vestitus*: *Cerianthus membranaceus* from GenBank (Appendix A) and original material of *P. solitarius* from Marcel (Appendix A). All our genetically graded material nested within *Synarachnactis lloydii* (Figure 1 and Figure 2, Appendix A), outside of the *Cerianthus membranaceus* clade, whereas Medditerranean *P. solitarius* was well placed within the *Pachycerianthus* spp. clade (Figure 1, Appendix A). *Synarachnactis lloydii* has probably existed in the Black Sea since its re-colonization ~7500–6500 years ago [54,55]. On the other hand, *Cerianthus lloydii* reported from the shallow (1–10 m) Aegean Sea [56] and the Sea of Marmara [57] may represent some other species, perhaps conspecific with *Cerianthus vestitus* (Forbes, 1843).

## 5. Conclusions

With newly generated data, we fully support the phylogenetic reconstruction by Forero Meija and co-authors [3] with better resolution of the previously unstudied clade A. From our dataset, it is possible to see that specimens originally determined as *Cerianthus lloydii* [9,11] are distant from the type species of the genus *Cerianthus (Cerianthus membranaceus*, clade B) and form a well-supported sister clade to the genus *Botrucnidifer*. Moreover, we confirm that the specimens previously reported as *Cerianthus lloydii* represent, in fact, several closely related species with patterns of distribution that probably reflect repeated invasions of Pacific fauna into the Arctic. To resolve the polyphyly of the genus *Cerianthus* we formally propose to resurrect the genus *Synarachnactis* Carlgren, 1924, to accommodate the species previously reported as *Cerianthus lloydii*. To reflect the new phylogeny, we propose a new family Synarachnactidae to accommodate the genus *Synarachnactis*. 

For the first time, *Synarachnactis lloydii* is reported in the Black Sea fauna list. We suggest that this species has been inhabiting the Black Sea since the Holocene re-colonization. 

## Figures and Tables

**Figure 1 biology-12-01167-f001:**
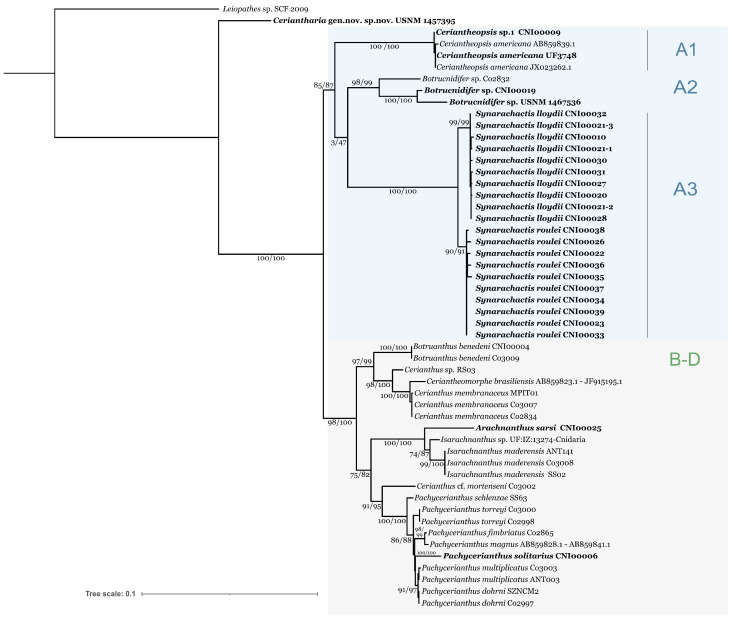
Concatenated 18S COI ML phylogenetic reconstruction. The SH-aLRT support (%)/ultrafast bootstrap support (%) are shown. Model selected: TIM2e+R2: 18S, TIM2e+G4: COI (1 position), F81+F+G4: COI (2 position), TPM2u+F+I+G4: COI (3 position). Data generated in our study are marked in bold. For clades explanation see 3.1.

**Figure 2 biology-12-01167-f002:**
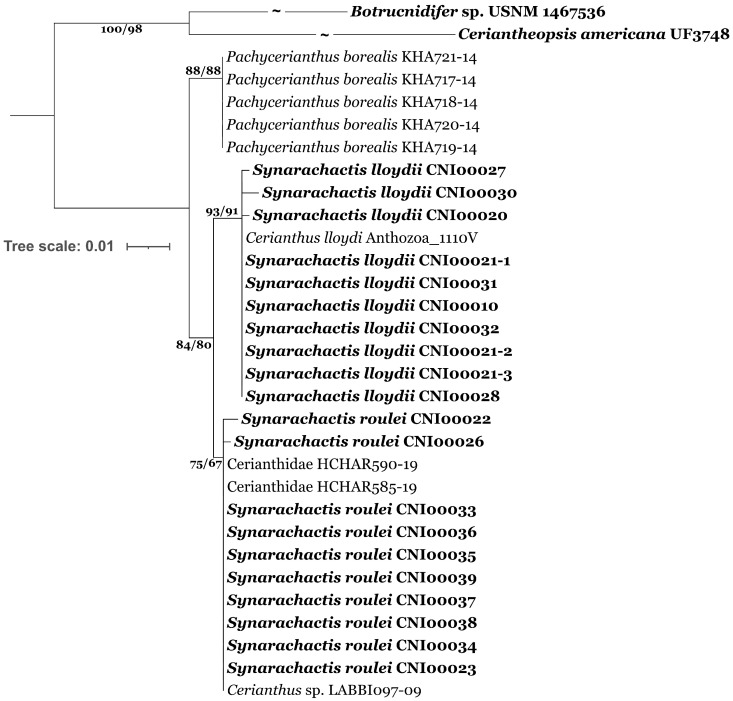
Phylogenetic reconstruction of A clade based on COI based on 591 bp alignment (ML). The SH-aLRT support (%)/ultrafast bootstrap support (%) are shown. Model selected: HKY+F:COI (1 position), K2P:COI (2 position), F81+F:COI (3 position). Data generated in our study are marked in bold.

**Figure 3 biology-12-01167-f003:**
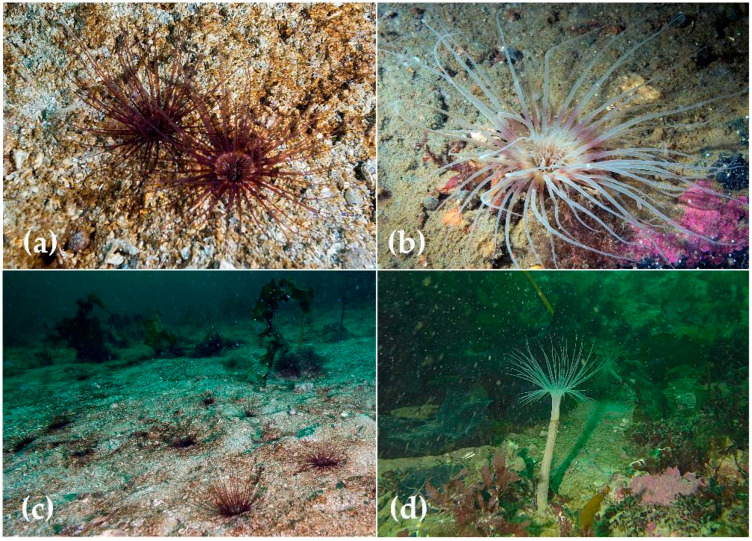
Underwater photographs of specimens from A3 clade: (**a**,**c**) *Synarachnactis lloydii* from the Dolgaya Bay, the Barents Sea; (**b**) *Synarachnactis roulei* from Cape Kindo, the White Sea. Photos T. Antokhina; (**d**) *Synarachnactis roulei* from Severnaya Zemlya, Akhmatova Bay; Photo Expedition “Open Ocean: Arctic Archipelagoes—2019. Severnaya Zemlya”.

**Figure 4 biology-12-01167-f004:**
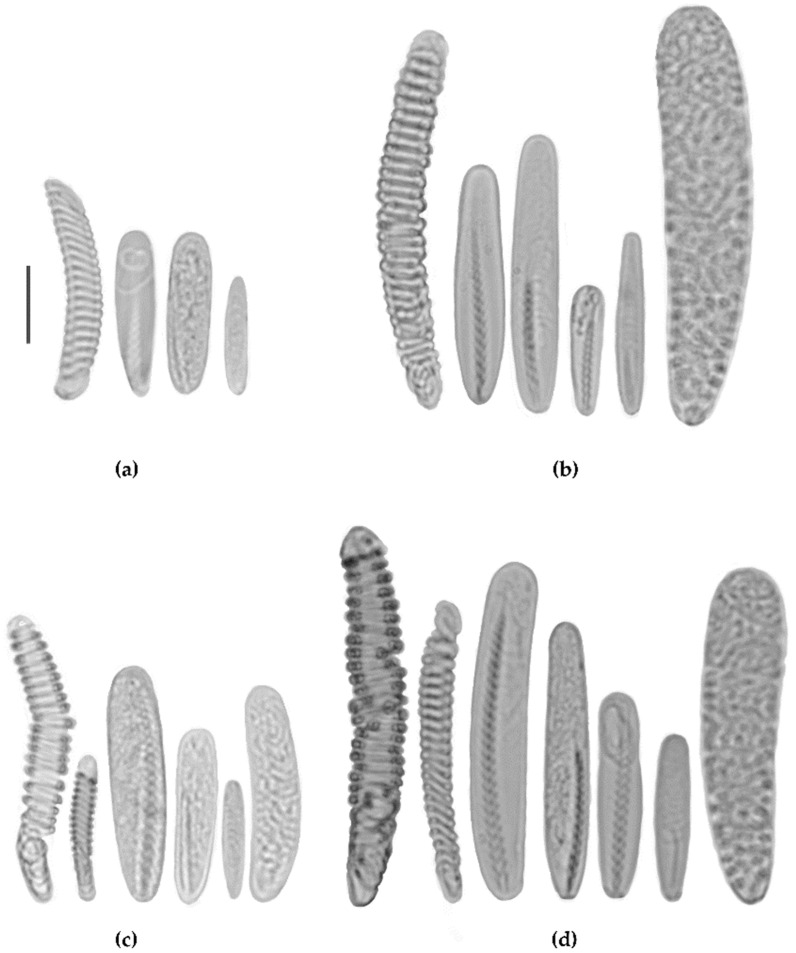
Cnidae from marginal (**a**,**b**) and labial (**c**,**d**) tentacles of *Synarachnactis lloydii* (IORAS CNI00010) (**a**,**c**) and *S. roulei* (IORAS CNI00038) (**b**,**d**). Scale 10 μm.

**Figure 5 biology-12-01167-f005:**
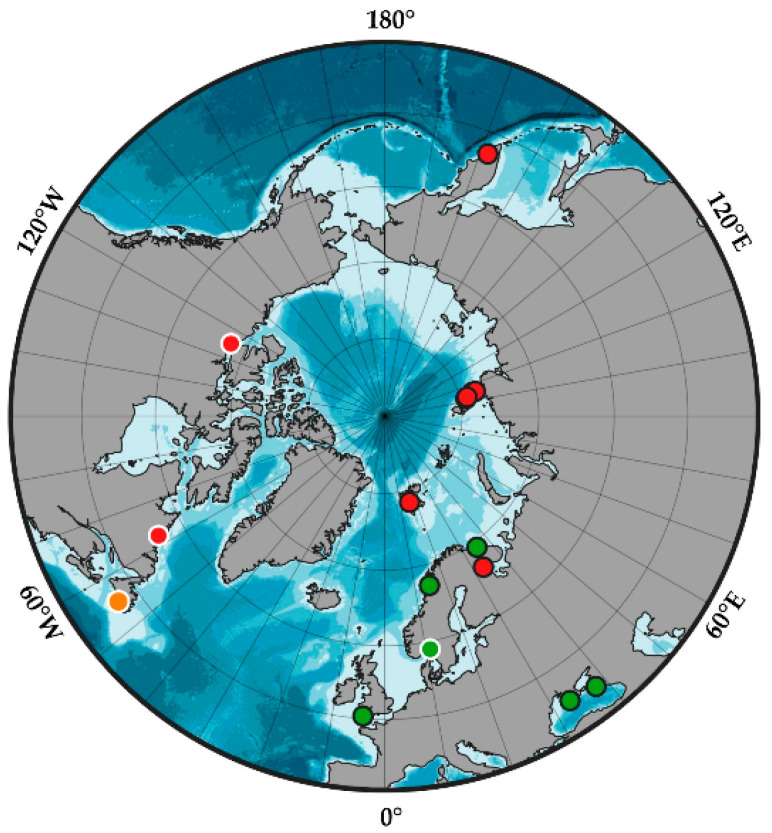
Geographical distribution of *Synarachnactis* spp. in the Northern Hemisphere according to analyzed genetic data. Green circles—*S. lloydii*; red circles—*S. roulei*; orange circles—*Synarachnactis* sp. (=*Pachycerianthus borealis* misindet.) from BOLD. Circles with black outline—original data; circles with white outline—BOLD data. For locations, see Appendix A.

**Table 1 biology-12-01167-t001:** Primers and annealing temperatures used in this study.

Gene	Primer	Direction	Sequence 5′–3′	PCR Scheme	Reference
COI	LCO1490	F	GGTCAACAAATCATAAAGATATTGG	at 95 °C for 15 s, annealing at 50 °C for 30 s, and extension at 72 °C for 45 s, for 5 cycles, at 95 °C for 15 s, annealing at 52 °C for 30 s, and extension at 72 °C for 45 s, for 30 cycles	[18]
	HCO2198	R	TAAACTTCAGGGTGACCAAAAAATCA		[18]
	dgLCO1490	F	GGTCAACAAATCATAAAGAYATYGG		[19]
	dgHCO2198	R	TAAACTTCAGGGTGACCAAARAAYCA		[19]
18S	TimA	F	AMCTGGTTGATCCTGCCAG	at 95 °C for 15 s, annealing at 55 °C for 30 s, and extension at 72 °C for 60 s for 35 cycles	[20]
	1100R2	R	CGGTATCTGATCGTCTTCGA		[20]
	18S–5F	F	GCGAAAGCATTTGCCAAGAA		[21]
	18S–9R	R	GATCCTTCCGCAGGTTCACCTAC		[21]

## Data Availability

The data generated in this study are openly available and deposited in the National Center for Biotechnology Information (NCBI) databank under the accession numbers: 18S-OR127166, OR127168, OR127175, OR127167, OR127158, OR127156, OR127157, OR127161, OR127162, OR127164, OR127165, OR127174, OR127153, OR127154, OR127155, OR127163, OR127159, OR127160, OR127169, OR127170, OR127171, OR127172, OR127173, OR127151, OR127152; COI-OR126090, OR126092, OR126099, OR126091, OR126081, OR126080, OR126083, OR126079, OR126085, OR126086, OR126088, OR126089, OR126098, OR126075, OR126076, OR126077, OR126078, OR126087, OR126082, OR126084, OR126093, OR126094, OR126095, OR126096, OR126097, OR126073, OR126074. Public data used in our study are accessible via NCBI and BOLD databases; see Appendix A for Genbank accession numbers and BOLD sequence IDs.

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
