# Peer review of "Cerianthus lloydii (Ceriantharia: Anthozoa: Cnidaria): New Status and New Perspectivesâ€"

_biology, 2023, doi:10.3390/biology12091167_

Round 1
Reviewer 1 Report
This paper reports on phylogenetic relationships among cerianthids and clarifies the status of the supposedly wide-spread species, Cerianthus lloydi. I believe the methods and results are sound. However, I think the paper can be improved by adding more detail and clarification in the discussion.
A major focus of the paper is reclassification of specimens previously identified as Cerianthus lloydi into two species with a resurrected genus: Synarachnactis lloydi and S. roulei. Good diagnoses are provided for the genus and family levels, but differences between these two species are only provided in a single paragraph and one figure. Illustrations are needed to support the claimed differences in b-rhabdoids. Updated diagnoses for each species need to be provided as well. I greatly appreciate that the authors included live photos of each species and showed their distributions on a polar projection map.
The comments on the Black Sea specimens are confusing. The paragraph begins with an assertion that S. lloydi’s occurrence is not a recent invasion, but then concludes by stating that the specimens were not studied in detail (based on previous misidentifications) and may be some other species. I feel like I’m missing a lot of details here, so the story just needs to be explained in more depth.
The manuscript is well-written and readable in English, but there are some minor grammatical mistakes in the writing. I would be willing to correct the revised manuscript if the authors could provide me with a Word document (not PDF). More importantly, the manuscript should be proof-read by the authors, because there are figure numbers missing in several cases. The number of specimens analyzed from the Florida Museum of Natural History is also missing from the main text. The “XX” designations must be replaced with the appropriate numbers prior to resubmission.
More minor comments are below:
I believe that diagnoses should have their own sub-headings within the discussion.
In Figure 1, the circles should be replaced with numbers indicating support. This would make the figure consistent with Fig. 2 and how most phylogenies are depicted.
What does it mean for a specimen to be bolded in Fig. 1? This should be explained in the caption.
The English requires minor editing for small grammatical mistakes. The manuscript is very readable but could benefit from that minor editing.
Author Response
First we would like to thank Reviewer1 for all comments and recommendations.
This paper reports on phylogenetic relationships among cerianthids and clarifies the status of the supposedly wide-spread species, Cerianthus lloydii. I believe the methods and results are sound. However, I think the paper can be improved by adding more detail and clarification in the discussion.
We modified some parts of the discussion section for clarification. Also we added more details requested by reviewer 1.
A major focus of the paper is reclassification of specimens previously identified as Cerianthus lloydi into two species with a resurrected genus: Synarachnactis lloydi and S. roulei. Good diagnoses are provided for the genus and family levels, but differences between these two species are only provided in a single paragraph and one figure.
- Illustrations are needed to support the claimed differences in b-rhabdoids.
we provide requested Figure (Figure 4 in the main text), that illustrates differences in nematocysts size and composition for marginal and labial tentacles of S. lloydii and S. roulei - Updated diagnoses for each species need to be provided as well.
We are not providing diagnoses for Synarachnactis spp. for a following reason. Most of the features used previously to distinguish Cerianthus lloydii from other Cerianthus appeared to have generic and even family level. The diagnosis of Cerianthus lloydii compiled by Oscar Carlgren (1912) that is generally used for identification of this species includes materials from Svalbard and the Kara Sea (apparently C.roulei). No morphological differences apart of larger size were provided by Carlgren. Cerianthus roulei in turn was named based on a sloppy and extremely over-stylized illustration that depicted a non-existing feature. No diagnosis was provided for this species. The study of the type material of C. roulei showed that the species is very closely related to Cerianthus lloydii morphologically. Much more material need to be analyzed to find any reliable morphological features that can be used for Diagnoses of these two species. Re-description of these two species and amended diagnoses will be published separately (we indicate it in the MS). - I greatly appreciate that the authors included live photos of each species and showed their distributions on a polar projection map.
We would like to thank reviewer for this kind remark.
- The comments on the Black Sea specimens are confusing. The paragraph begins with an assertion that S. lloydi’s occurrence is not a recent invasion, but then concludes by stating that the specimens were not studied in detail (based on previous misidentifications) and may be some other species. I feel like I’m missing a lot of details here, so the story just needs to be explained in more depth.
The Black Sea cerianthid was never studied in details before our work. The name used in local faunal reports was never questioned as it was the only cerianthid in the Black sea, known from very characteristic biotope. We thank reviewer for this comment and we did our best to revise this part of the MS to make all our statements clearer.
- The manuscript is well-written and readable in English, but there are some minor grammatical mistakes in the writing. I would be willing to correct the revised manuscript if the authors could provide me with a Word document (not PDF). More importantly, the manuscript should be proof-read by the authors, because there are figure numbers missing in several cases.
We are extremelly grateful to reviewer 1 for this comment, however, as we discovered from the handling editor MDPI Biology does not provide name of reviewer till the moment the MS is published. Thus, we cannot use this generous offer. However, we did our best to correct English.
- The number of specimens analyzed from the Florida Museum of Natural History is also missing from the main text. The “XX” designations must be replaced with the appropriate numbers prior to resubmission.
Corrected
More minor comments are below:
- I believe that diagnoses should have their own sub-headings within the discussion.
We have sub-titles for the genus (4.2.1 Synarachnactis Carlgren 1924 (amended)) and family (4.2.2 Synarachnactidae fam. nov. Molodtsova & Simakova, 2023). According to Biology guidelines we cannot use further subtitles. We will put a request to highlight Diagnoses with bold or Italics - In Figure 1, the circles should be replaced with numbers indicating support. This would make the figure consistent with Fig. 2 and how most phylogenies are depicted.
new version of figure 1 with numbers indicating support is provided.
- What does it mean for a specimen to be bolded in Fig. 1? This should be explained in the caption.
This comment is highly appreciated We have lost a line when formatted the MS. Bold was used to highlight data generated in our study. We added this to Figures 1 and 2 captions.
Quality of English Language
The English requires minor editing for small grammatical mistakes. The manuscript is very readable but could benefit from that minor editing.
we did our best to correct English
Reviewer 2 Report
The authors do a good job of describing the study in the introduction. Just the right amount of factual information, nothing extraneous. In fact, the whole paper is well-written, clear, and informative. The methods are thoroughly described, so that another could easily follow the protocols. This is unusual in papers these days, which leave much of the specifics up to the imagination. Table 1 is excellent.
I believe the analyses and results to be complete and not over-complicated. It seems to me that the data has been analyzed using current best practices. A dataset consisting of only two loci is less common these days, but I think the data collected is sufficient to support the conclusions.
I am unfamiliar with these taxa, but the genus description seems clear and sufficient for identification.
I believe the very last sentence could be rewritten for greater clarity.
Other than a few instances of XX when a number is needed, I find the paper to be a good example of its type.
Throughout the paper, the articles "the," "a," and "an" are missing and should be inserted.
Author Response
We would like to thank Reviewer2 for all comments provided
The authors do a good job of describing the study in the introduction. Just the right amount of factual information, nothing extraneous. In fact, the whole paper is well-written, clear, and informative. The methods are thoroughly described, so that another could easily follow the protocols. This is unusual in papers these days, which leave much of the specifics up to the imagination. Table 1 is excellent.
I believe the analyses and results to be complete and not over-complicated. It seems to me that the data has been analyzed using current best practices. A dataset consisting of only two loci is less common these days, but I think the data collected is sufficient to support the conclusions.
I am unfamiliar with these taxa, but the genus description seems clear and sufficient for identification.
We would like to thank reviewer 2 for these comments.
I believe the very last sentence could be rewritten for greater clarity.
We rephrase this sentence as following: “For the first time Synarachnactis lloydii is reported in the Black Sea fauna list. We suggest that this species has been inhabiting the Black Sea since the Holocene re-colonization.”
Other than a few instances of XX when a number is needed, I find the paper to be a good example of its type.
All numbers are added
Round 2
Reviewer 1 Report
The authors have responded to all my original comments in a satisfactory way. I think the manuscript is improved, particularly the clarity in the discussion. I have no further suggested edits.
Author Response
We would like to thank reviewer 1 for comments and observations that allowed us to improve the MS